# Collagen Peptides from Swim Bladders of Giant Croaker (*Nibea japonica*) and Their Protective Effects against H_2_O_2_-Induced Oxidative Damage toward Human Umbilical Vein Endothelial Cells

**DOI:** 10.3390/md18080430

**Published:** 2020-08-18

**Authors:** Jiawen Zheng, Xiaoxiao Tian, Baogui Xu, Falei Yuan, Jianfang Gong, Zuisu Yang

**Affiliations:** 1Zhejiang Provincial Engineering Technology Research Center of Marine Biomedical Products, School of Food and Pharmacy, Zhejiang Ocean University, Zhoushan 316022, China; jwzheng1996@163.com (J.Z.); TIANXIAOXIAO0208@163.com (X.T.); xubaogui96@163.com (B.X.); yuanfalei@zjou.edu.cn (F.Y.); 2Donghai Science and Technology College, Zhejiang Ocean University, Zhoushan 316000, China; gjf200527@163.com

**Keywords:** *Nibea japonica*, swim bladder, marine collagen peptides, antioxidant activity

## Abstract

Five different proteases were used to hydrolyze the swim bladders of *Nibea japonica* and the hydrolysate treated by neutrase (collagen peptide named SNNHs) showed the highest DPPH radical scavenging activity. The extraction process of SNNHs was optimized by response surface methodology, and the optimal conditions were as follows: a temperature of 47.2 °C, a pH of 7.3 and an enzyme concentration of 1100 U/g, which resulted in the maximum DPPH clearance rate of 95.44%. Peptides with a Mw of less than 1 kDa (SNNH-1) were obtained by ultrafiltration, and exhibited good scavenging activity for hydroxyl radicals, ABTS radicals and superoxide anion radicals. Furthermore, SNNH-1 significantly promoted the proliferation of HUVECs, and the protective effect of SNNH-1 against oxidative damage of H_2_O_2_-induced HUVECs was investigated. The results indicated that all groups receiving SNNH-1 pretreatment showed an increase in GSH-Px, SOD, and CAT activities compared with the model group. In addition, SNNH-1 pretreatment reduced the levels of ROS and MDA in HUVECs with H_2_O_2_-induced oxidative damage. These results indicate that collagen peptides from swim bladders of *Nibea japonica* can significantly reduce the oxidative stress damage caused by H_2_O_2_ in HUVECs and provides a basis for the application of collagen peptides in the food industry, pharmaceuticals, and cosmetics.

## 1. Introduction

Redox reactions are basic physiological and chemical reactions that are constantly carried out in the human body. During metabolism in the human body, large amounts of reactive oxygen species (ROS) are generated and the antioxidant defense system in the body removes active oxygen free radicals in order to prevent damage to the body [1,2]. Under normal circumstances, a healthy human body has an efficient and dynamic antioxidant defense system where the generation and removal of oxidative radicals in the human body are in a dynamic balance. However, once the human body generates excessive reactive oxygen radicals or the antioxidant defense system becomes inefficient, this dynamic balance is disturbed, which generates a state of oxidative stress, leading to aging and a number of chronic diseases, such as diabetes and coronary arteriosclerosis [1,3]. ROS are the products of redox reactions in the body and are oxygen free radicals. In addition, ROS play direct, critical roles in the body’s oxidative stress reaction and are one of the most harmful free radicals to human tissues. Furthermore, ROS can produce excessive oxidation on cell membrane structures, nucleic acids, lipids and proteins as well as cause changes in tissue structure and produce adverse effects, such as physiological dysfunction [4,5,6]. Under normal circumstances, the body can remove excess ROS in two ways to reduce the damage caused by oxidative stress. One way is through a series of antioxidant mechanisms in the body, which clear excess ROS and maintain the dynamic balance of ROS. These antioxidant mechanisms involve glutathione peroxidase (GSH-Px), superoxide dismutase (SOD), catalase (CAT) and other small-molecule reducing substances. The second way is the intake of external substances, such as vitamin C, carotenoids, glutathione (GSH) and melatonin [7,8]. Therefore, supplementing with antioxidants to eliminate excess ROS may be an effective means for treating chronic diseases [7].

In addition, antioxidant peptides prevent peroxidation in the body and help the body clear active oxygen free radicals [9]. In recent years, an increasing number of studies have found that certain peptides have the ability to scavenge free radicals, which provides a new idea for identifying novel antioxidants. However, there are only a few types of naturally-occurring, antioxidant-active peptides, which has made obtaining novel antioxidant-active peptides by hydrolyzing macromolecular proteins a research hotspot [10,11]. In addition, research related to food health products, cosmetics, and pharmaceutical industries with antioxidants as the mechanism will also move in the direction of researching active antioxidant peptides [12]. For example, Chi et al. [13] obtained three antioxidant peptides from the protein hydrolysate of bluefin leatherjacket (*Navodon septentrionalis*) skin, which all had molecular weights (Mw) of less than 700 Da. In addition, the strength of the antioxidant activities of these peptides increased as their Mw decreased. Another example is a small-molecule antioxidant peptide obtained from horse mackerel (*Magalaspis cordyla*) viscera protein. This antioxidant peptide had a Mw of 518.5 Da and had a good ability to scavenge 2, 2-diphenyl-1-picrylhydrazyl (DPPH) and hydroxyl radicals [14]. You et al. [15,16,17] hydrolyzed loach, jellyfish and grass carp. After these hydrolyzed polypeptide products were administered to mice over time by oral gavage, the activity of the antioxidant enzyme system, SOD and GSH-Px, was detected to varying degrees. Taken together, these results indicate that marine proteins are a high-quality raw material for preparing antioxidant peptides.

Previously, acid soluble collagen (ASC) and pepsin soluble collagen (PSC) were extracted from the swim bladders and skin of *Nibea japonica*. Both ASC and PSC are type I collagens, which have potential wound healing functions and can be applied to the field of cosmetics [18,19,20]. However, there have been no reports on the extraction of antioxidant peptides from the swim bladders of *Nibea japonica*. Therefore, our study further utilizes the swim bladders of *Nibea japonica* and used neutrase to obtain products with the highest DPPH free radical scavenging rate (SNNHs). After ultrafiltration, the hydrolysate with Mw of less than 1 kDa (SNNH-1) had the highest scavenging rate and was extracted for functional evaluation. Here, the antioxidant activity of SNNH-1 in vitro was analyzed. Our findings showed that SNNH-1 can be used as a marine antioxidant and provide a basis for its application in the food and pharmaceutical fields.

## 2. Results and Discussion

### 2.1. Single Factor Results

#### 2.1.1. Selection of the Optimal Enzyme

To obtain the hydrolysate with ideal activity, research must be carried out to find the best proteolytic enzyme. The defatted swim bladders were used to hydrolyze the protein for 4 h at the optimal temperature and pH of each enzyme (trypsin at pH 8.0, 50 °C; neutrase at pH 7.0, 45 °C; alcalase at pH 9.0, 50 °C; pepsin at pH 2.0, 37 °C; and papain at pH 6.0, 55 °C). Each enzyme was used at 1000 U/g. The swim bladders were mixed with water at the ratio of 1:4. To inactivate the enzyme, the hydrolysate was boiled for 10 min. Subsequently, the hydrolysate was centrifuged at 12,000× *g* rpm for 10 min at 4 °C. The supernatant was stored at −80 °C overnight, freeze-dried, and stored at −20 °C. The optimal hydrolase of the swim bladders of *Nibea japonica* was screened by the DPPH clearance rate.

As shown in Figure 1, the neutrase hydrolysate exhibited the highest DPPH radical scavenging activity. Therefore, neutrase was selected for the preparation of proteolytic products of the swim bladder from *Nibea japonica*. Neutrase hydrolysate obtained under the optimum conditions was called SNNHs and was stored at −20 °C.

#### 2.1.2. Single Factor Experiments

Figure 2 shows the influence of five single factors on DPPH radical scavenging activity. Within the range of the five single factors, the DPPH clearance rate was first increased to the maximum value and then decreased. The optimal enzymolysis conditions corresponding to the highest level of DPPH clearance in the range test were as follows: enzyme concentration of 1000 U/g, a solid:liquid ratio of 1:5 (*w*/*v*), a hydrolysis time of 6.0 h, a pH of 7.0, and a temperature of 45 °C.

### 2.2. Optimization of Extraction Parameters by Response Surface Methodology (RSM)

#### 2.2.1. Response Surface Analysis

According to the results of the single factor experiment, the RSM of the Box–Benhnken (BBD) was used to analyze the optimal levels of three independent factors that had significant influences on DPPH radical scavenging rate. The experimental design and results are listed in Table 1. Based on the regression analysis of the data, the second-order polynomial function was used to predict the effect of these three factors on DPPH clearance, as follows: *Y* = 94.16 + 3.37*X*_1_ + 1.97*X*_2_ + 1.47*X*_3_ − 0.47*X*_1_*X*_2_ + 0.14*X*_1_*X*_3_ − 1.18*X*_2_*X*_3_ − 3.68*X*_1_^2^ − 2.34*X_2_*^2^ − 1.79*X*_3_^2^ (where *Y* was the DPPH clearance rate, and *X*_1_, *X*_2_ and *X*_3_ were the temperature, pH, and enzyme concentration, respectively).

Table 2 presents the analysis of variance (ANOVA) results of the quadratic model, from which the significance of the quadratic model can be determined. When the value of *F* was larger and the value of *p* was smaller, the corresponding variable was more statistically significant. If the *p* value was greater than 0.05, then the model item was not statistically significant. The *F*-value and *p*-value (*p* = 0.0004) demonstrate that the model had high significance. The model’s ANOVA decision coefficient (*R*^2^ = 0.9860) and adjusted decision coefficient (*R_Adj_*^2^ = 0.9615) also show that the model was highly significant. Therefore, we decided on this model for optimization.

In addition, three-dimensional response surfaces and contour plots generated by the model equation can intuitively explain the interaction between the two factors (Figure 3) and show the optimal levels of each component required for the maximum DPPH clearance rate. The maximum DPPH radical scavenging rate was 95.44% at the following conditions: temperature was 47.2 °C, pH was 7.3, and enzyme concentration was 1100 U/g.

#### 2.2.2. Validation of the Models

Experiments were performed in triplicate and were carried out under the optimal extraction conditions: temperature was 47.2 °C, pH was 7.3, and enzyme concentration was 1100 U/g. Under similar conditions, the average DPPH radical scavenging rate was 94.85%, which was in good agreement with the predicted value.

### 2.3. Molecular Weight Distribution of SNNHs

Antioxidant peptides with different Mw have different free radical scavenging capabilities. To determine the Mw distribution of SNNHs, taking the retention time (*Rt*) as the abscissa and log(*Mw*) as the ordinate, the following regression equation was obtained: log(*Mw*) = − 0.477*Rt* + 7.0117. The value of the measurement coefficient (*R*^2^) was 0.9996, which showed a good linear relationship. This indicated that the relative Mw of the SNNHs could be analyzed based on this linear regression equation (Figure 4A).

The HPLC gel filtration chromatogram of SNNHs of *Nibea japonica* under the same chromatographic conditions is shown in Figure 4B. The components of less than 1, 1–5, 5–10 and more than 10 kDa accounted for 47.88%, 46.21% 1.04% and 4.76% of the components, respectively [21], which showed that the SNNHs consisted mainly of many small peptides. SNNHs were more water-soluble and antioxidant than the collagen found in the swim bladders of *Nibea japonica*.

### 2.4. Fractionation of SNNH-1 from SNNHs

Based on the principle of mechanical retention, the enzymatic hydrolysates in the swim bladders of *Nibea japonica* were isolated by ultrafiltration membranes and divided into four parts with different Mw distributions. Subsequently, the DPPH radical scavenging rate of each part was determined. As shown in Figure 5, peptide fractions of less than 1 kDa had the highest DPPH free radical scavenging rate compared to the other ultrafiltration fractions. Similar research has shown that the low-Mw content of protein hydrolysates has higher antioxidant activity [22]. Therefore, fractions with a Mw of less than 1 kDa were chosen for subsequent activity evaluation and named SNNH-1.

### 2.5. Amino Acid Composition of SNNH-1

The antioxidant properties of peptides are related to their amino acid composition [23]. The acidic and basic amino acids of the peptide segment help antioxidant peptides obtain better metal ion chelating and free radical scavenging abilities [24]. In addition, peptide chains containing hydrophobic amino acids can better exist at the water–lipid interface, which improves the free radical scavenging ability of the polypeptide [25]. Therefore, the amino acid composition of SNNH-1 was analyzed and the results were expressed in residues per 1000 total residues. Table 3 shows that SNNH-1 contained 7 essential amino acids and 10 non-essential amino acids. The highest amino acid content of SNNH-1 was glycine, alanine, proline, and hydroxyproline, which accounted for 19.23%, 13.34%, 11.87%, and 10.28% of the amino acid content, respectively. The amino acid content of SNNH-1 products was consistent with the collagen peptides from *Nibea japonica* in a previous study (glycine (21.22%), alanine (9.79%), proline (10.78%) and hydroxyproline (9.28%)) [26]. In addition, cysteine was not detected in SNNH-1. It has previously been reported that the activity of the polypeptide formed after protein hydrolysis is related to the content of hydrophobic amino acids contained in the polypeptide [27]. For example, a polypeptide with antioxidant activity has mostly N-terminal hydrophobic amino acids, and the presence of hydrophobic amino acids is positively correlated with its antioxidant activity [28]. The content of the hydrophobic amino acids, proline and alanine, in SNNH-1 was 11.87% and 13.34%, respectively, which is relatively high and suggests that SNNH-1 has strong antioxidant activity.

### 2.6. Antioxidant Activity of SNNH-1

Free radicals can damage biological membranes because the excessively high oxygen concentration in the lipid bilayer of the membrane is easily attacked by free radicals, which results in lipid peroxidation. Free radicals generated during the process of lipid peroxidation inactivate the transport enzymes on the membrane, result in an imbalance between the internal and external environment of the cell, reduce the fluidity of the membrane, and eventually cause damage to the entire organization and function of the cell [2,29,30].

Radical scavenging activity is a significant interest for the cosmeceutical industry in order to prevent photoaging and ultraviolet damage. To assess the antioxidant activity of SNNH-1, assays for DPPH radical, hydroxyl radical, superoxide anion radical, and 2,2′-azino-bis-3-ethylbenzothiazoline-6-sulfonic acid (ABTS) radical scavenging were used and compared with GSH as an activity control. As shown in Figure 6, SNNH-1 from the swim bladders of *Nibea japonica* has a dose-dependent scavenging effect on DPPH radicals, hydroxyl radicals, superoxide anion radicals and ABTS radicals. SNNH-1 showed high scavenging capabilities of DPPH radicals (Figure 6A), hydroxyl radicals (Figure 6B), ABTS radicals (Figure 6C) and superoxide anion radicals (Figure 6D). Furthermore, the antioxidant activity of SNNH-1 is close to that of GSH. Therefore, the high free radical scavenging activity of SNNH-1 indicates that it is a potential candidate to be developed as an antioxidant and can be used in anti-aging health products and cosmetics.

### 2.7. Cytotoxic and Allergenic Potential of SNNH-1

The cytotoxicity and allergenic potential of SNNH-1 were assessed by the MTT method and lactate dehydrogenase (LDH) toxicity test, respectively. As shown in Figure 7A, human umbilical vein endothelial cells (HUVECs) treated with different concentrations of SNNH-1 for 24 h did not show decreased viability. In contrast, SNNH-1 promoted the growth of HUVECs. Therefore, SNNH-1 has no obvious cytotoxic effects in vitro. LDH is a glycolytic enzyme that is widely present in the cytoplasm of cells. Normally, LDH is only in the cytoplasm and measuring LDH levels in cell culture supernatants is a sensitive indicator of cell damage. Increased LDH levels in cell culture supernatants indicates that the cells have been damaged, and that the membrane permeability of the cells has increased. Therefore, the amount of LDH that has leaked out of the membrane from the cytoplasm reflects the degree of cell damage [31]. In addition, the release of LDH can be closely related to allergic reactions and inflammation. In this study, LDH release from SNNH-1-treated cells was lower than that from untreated cells (Figure 7B). These results were consistent with the data presented by Yang et al. [26], who showed that MCPs from *Nibea japonica* skin have the potential to promote cell growth. Therefore, SNNH-1 may be considered as a non-cytotoxic and hypoallergenic material.

### 2.8. Effects of SNNH-1 on the Levels of GSH-Px, SOD, CAT and Malondialdehyde (MDA) in an H_2_O_2_-Induced HUVECs Injury Model

To evaluate the antioxidant activity of SNNH-1, we investigated the effect of SNNH-1 pretreatment on the levels of GSH-Px, SOD, CAT and MDA after oxidative damage of HUVECs induced by H_2_O_2_. HUVECs have various functions and are often used to study the relationship between cardiovascular disease and oxygen-free radicals. Therefore, HUVECs were selected as the cell model of oxidative stress in this experiment [6]. H_2_O_2_ injury is currently the most widely used cell injury model. Previous studies have shown that H_2_O_2_ can not only attack biofilms and trigger lipid peroxidation reactions, which destroys the integrity of biofilms, but it can also reduce the activity of antioxidant enzymes in cells [32,33]. CAT, SOD and GSH-Px are all important components of enzymes in the cell’s antioxidant defense system, and they have very important impacts on the body’s oxidation and antioxidant homeostasis. CAT can decompose H_2_O_2_ into water and oxygen [34]. SOD scavenges superoxide anion free radicals and prevents cell damage. GSH-Px can promote the reaction of hydrogen peroxide and GSH to produce water and oxidized glutathione (GSSG). As shown in Figure 8A–C, H_2_O_2_ treatment significantly reduced the activities of GSH-Px, SOD and CAT in HUVECs. However, in the SNNH-1 pretreated group, the activities of these three enzymes were increased significantly in a dose-dependent manner. These data show that SNNH-1 inhibits intracellular lipid peroxidation to a certain extent and enhances the cell’s antioxidant defense system.

MDA is a product of peroxidized lipids and can attack unsaturated fatty acids in the cell membrane, which causes cell damage. Therefore, the degree of lipid peroxidation and cell damage can be judged according to the MDA content in cells [35]. As shown in Figure 8D, the MDA content in the cells was significantly higher after H_2_O_2_ treatment compared to the control group, which indicates that HUVECs were damaged by H_2_O_2_ and a large amount of MDA was formed. However, pretreatment of HUVECs with SNNH-1 decreased the amount of MDA with increasing peptide concentrations resulting in lower MDA contents. Furthermore, the MDA level in the high-dose group was similar to the MDA amount in the control group. In addition, Cai et al. [36] reported that FPYLRH (S8) from the swim bladders of Miiuy Croaker (*Miichthys miiuy*) can up-regulate the levels of SOD and GSH-Px, and down-regulate the contents of MDA, suggesting that it plays a protective role in the antioxidant effects on HUVECs against H_2_O_2_-induced injury.

### 2.9. Effects of SNNH-1 on ROS Levels in a H_2_O_2_-Induced HUVECs Injury Model

When the strength of the redox reaction inside the body is beyond the capacity of the body to resist oxidation, then the production of ROS will increase [32,37]. In previous studies, it was shown that H_2_O_2_ can increase the levels of ROS in cells. In addition, cells in an environment containing free radicals for a long time may cause damage to important biological macromolecules, DNA mutations, damage to organs and tissues, and diseases [35,38]. Vascular endothelial cells are one of the main sources of ROS in the body and these cells are involved in the process of disease. At each stage of disease formation, ROS have a direct impact. Therefore, detecting the content of ROS in cells can directly reflect the antioxidant capacity of the cells, as well as the degree of oxidative damage. Here, the effect of SNNH-1 on ROS levels in HUVECs was studied. As shown in Figure 9A,B, after H_2_O_2_ treatment, the fluorescence intensity in HUVECs was significantly higher compared to the control group. However, SNNH-1 pretreatment effectively reduced ROS levels in HUVECs. In addition, ROS levels decreased with increasing concentrations of SNNH-1. Furthermore, the level of H_2_O_2_ produced in HUVECs showed a similar result (Figure 9C). These results indicate that the protective effect of SNNH-1 on H_2_O_2_-induced HUVECs injury may be due to the inhibition of intracellular ROS production. These results are in accordance with the data presented by Li et al. [22], who found that collagen peptides from sea cucumbers (*Acaudina molpadioides*) could effectively protect cells from H_2_O_2_-induced damage.

## 3. Materials and Methods

### 3.1. Materials and Chemicals

Swim bladders of *Nibea japonica* were obtained from the Zhejiang Marine Fisheries Research Institution (Zhoushan, China). Five proteases (trypsin (≥250 U/mg), neutrase (≥60,000 U/g), alcalase (≥200 U/mg), pepsin (≥3000 U/mg) and papain (≥500 U/mg)) were purchased from Beijing Asia Pacific Hengxin Co., Ltd. (Beijing, China). DPPH, ABTS, and phenazine methosulfate (PMS) were purchased from Sigma Chemicals (Shanghai, China) Trading Co., Ltd. HUVECs were purchased from the Cell Bank of Type Culture Collection of the Chinese Academy of Sciences (Shanghai, China). All chemicals were of analytical grade.

### 3.2. Optimization of Preparative Conditions

The swim bladders of *Nibea japonica* were pretreated by the method presented by Chen et al. [20]. Five major factors were selected in the single-factor experiments to set up preliminary ranges of the extraction variables, which include temperature, hydrolysis time, pH, solid:liquid ratio and enzyme concentration.

To further optimize the extraction conditions of the antioxidant peptides of the swim bladder, according to the single factor experiments, the two least influential factors were eliminated. Using the three other factors (extraction temperature, pH, and enzyme concentration), a three-level and three-factor response surface test was designed through the BBD and Design Expert. The activity of the extracted antioxidant peptides was evaluated by DPPH clearance rate. According to preliminary experimental results, the range and level of independent variables (Table 4) were classified.

The BBD in the experiment design contained 15 experimental points (Table 2) and a multiple regression analysis was performed on the response obtained from each experimental design to fit the following quadratic polynomial model:(1)$$\gamma =\beta_{0}+\overset{k}{\underset{i=1}{\sum }}\beta_{i}X_{i}+\overset{k}{\underset{i=1}{\sum }}\beta_{ii}X_{i}^{2}+\sum \underset{i<j}{\sum }\beta_{ij}X_{i}X_{j}$$
where *γ* is the predicted response, *β*_0_ is the intercept, *β_i_*, *β_ii_* and *β_ij_* are the linear, quadratic, and interaction coefficients, respectively, and both *X_i_* and *X_j_* are independent factors. Each experimental design was performed in triplicate. According to Design Expert 8.0.6, a variance table analysis was generated to determine the influence of the regression coefficients on the linear, quadratic and interaction terms.

### 3.3. Determination of the Mw Distribution of SNNHs

The Mw distribution of SNNHs was analyzed by gel filtration chromatography using a high-performance liquid chromatography system (Agilent 1260, Palo Alto, CA, USA). Water:acetonitrile:trifluoroacetic acid (55:45:0.1) were adopted as the mobile phase, the flow rate was 0.5 mL/min and the UV wavelength was 220 nm [21]. The column was calibrated with standard materials: peroxidase (40,500 Da), aprotinin (6512 Da), insulin (5807 Da), *Cyclina sinensis* polypeptide (Arg-Val-Ala-Pro-Glu-Glu-His-Pro-Val-Glu-Gly-Arg-Tyl-Leu-Val, 1751.78 Da) [39], and *Anthopleura anjunae* oligopeptide (Tyr-Val-Pro-Gly-Pro, 531.61 Da) [40]. A standard curve of the log (*Mw*) with retention time was created. The Mw of SNNHs was calculated based on the retention time using the standard curve equation.

### 3.4. Fractionation of SNNH-1 by Ultrafiltration

A GM-18 Roll film separation system (Bona Biotechnology Co., Ltd., Jinan, China) was used to purify peptides with the highest DPPH free radical scavenging activity with 10, 5 and 1 kDa Mw cut-off membranes. Collected peptides with different Mw were separated and evaluated for their antioxidant activity after lyophilization [41].

### 3.5. Amino Acid Composition Measurement of SNNH-1

Amino acid analysis was determined based on the method described by Tang et al. [19]. In brief, SNNH-1 was hydrolyzed by dissolving it in 6 M HCl at 110 °C for 24 h without oxygen and then it was vaporized. The hydrolysate was analyzed by a Hitachi L-8800 amino acid analyzer (Hitachi, Tokyo, Japan).

### 3.6. Antioxidant Activity of SNNH-1

The DPPH, hydroxyl, ABTS and superoxide anion radical scavenging activity of SNNH-1 was performed as in the methods described by Chen et al. [20].

### 3.7. Cytotoxic and Allergenic Properties of SNNH-1

The cytotoxic and allergen properties of SNNH-1 were determined using HUVECs in accordance with the manufacturers’ instructions of the MTT assay kit and the LDH release assay. The MTT and LDH test methods were performed as described by Lin et al. [42].

### 3.8. Assays for Antioxidant Enzymatic Activity of SNNH-1 in H_2_O_2_-Induced HUVECs

HUVECs were inoculated into 6-well plates (1 × 10^5^ cells/well) and incubated in a 5% CO_2_ incubator at 37 °C for 24 h. SNNH-1 at final concentrations of 0, 25, 50 and 100 µg/mL were added into the protection groups and cultured for another 24 h. Each group was treated with 600 µmol/L H_2_O_2_ for 4 h. The group treated without SNNH-1 and H_2_O_2_ was used as the control group. Subsequently, 500 μL of cell lysis buffer was added to each well on ice. Cells were lysed for 30 min, and centrifuged at 12,000 rpm for 10 min at 4 °C. The resulting supernatant was stored at 4 °C. Levels of GSH-Px, SOD, CAT, MDA, and H_2_O_2_ were determined by using assay kits according to the manufacturers’ instructions (Nanjing Jiancheng Bioengineering Institute, Nanjing, China), and protein concentrations were determined using the bicinchoninic acid (BCA) method.

### 3.9. Determination of the Levels of ROS in H_2_O_2_-Induced HUVECs

HUVECs were seeded in 6-well plates with a density of 1 × 10^4^ cells/mL for 24 h. SNNH-1 at final concentrations of 25, 50 and 100 µg/mL was added to the protection groups for 24 h and HUVECs were treated with 600 µmol/L H_2_O_2_ for 4 h in a 5% CO2 incubator at 37 °C. Production of ROS was determined using a ROS Assay Kit (Nanjing Jiancheng Bioengineering Institute, Nanjing, China). Next, a total of 2 mL of 2′,7′-dichlorodihydrofluorescein diacetate (DCFH-DA) fluorescent probe solution (10 µM) was added to the cells, and incubated for 30 min. Then cells were washed three times with serum-free Dulbecco’s modified Eagle medium (DMEM) and observed under a fluorescence microscope (Axio Imager A2, Carl Zeiss, Oberkochen, Germany). The fluorescence intensity was analyzed using Image J software.

### 3.10. Statistical Analysis

Data are presented as the mean ± SD (*n* = 3). Multiple-group comparisons were determined using ANOVA (SPSS 19.0 software, Armonk, New York, NY, USA). *p* < 0.05 was considered statistically significant.

## 4. Conclusions

In this work, RSM was used to optimize the extraction process of SNNHs from the swim bladders of *Nibea japonica*. It was found that the optimal extraction conditions were a neutrase concentration of 1100 U/g, a temperature of 47.2 °C and a pH of 7.3, which resulted in the maximum DPPH clearance rate of 95.44%. SNNH-1 (Mw < 1 kDa) was obtained from the extracted collagen peptide SNNHs by ultrafiltration. SNNH-1 had good scavenging activities of DPPH, hydroxyl, ABTS, and superoxide anion radicals. In addition, SNNH-1 showed an important protective effect against H_2_O_2_ injury in HUVECs by promoting cell proliferation, reducing the contents of ROS and MDA, and enhancing the activity of antioxidant enzymes (GSH-Px, SOD, and CAT). These results provide a basis for the future application of SNNH-1 in food processing, pharmaceuticals and cosmetics.

## Figures and Tables

**Figure 1 marinedrugs-18-00430-f001:**
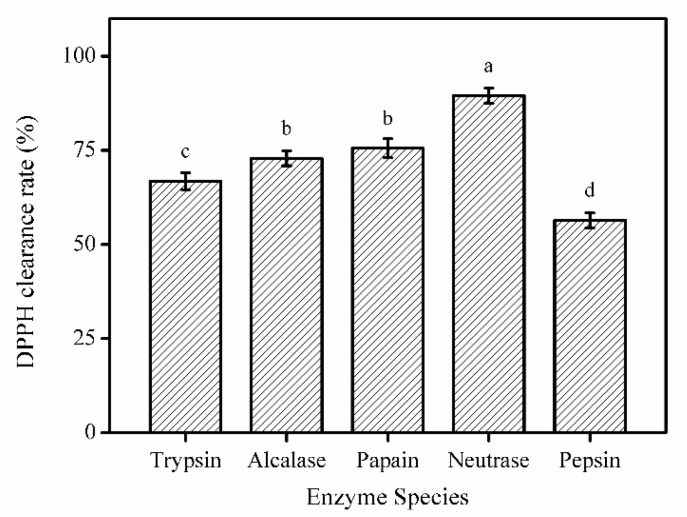
DPPH radical scavenging activity of hydrolysates produced by various proteases. The concentration of hydrolyzed product was 10 mg/mL. All results were triplicates of the mean ± SD. Different letters indicate significant differences between groups (*p* < 0.05).

**Figure 2 marinedrugs-18-00430-f002:**
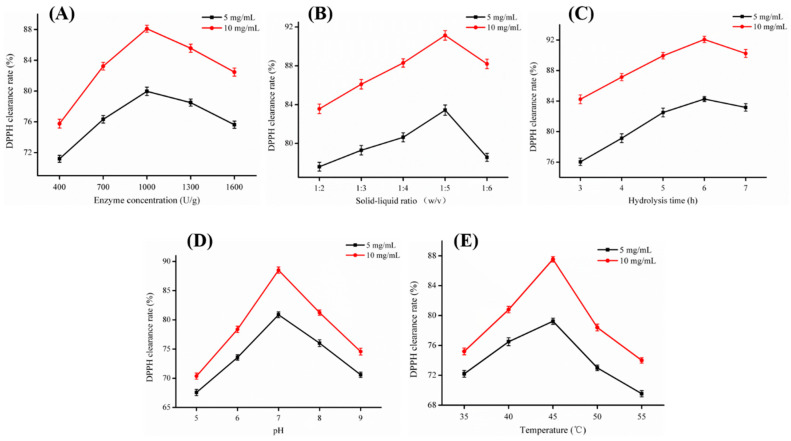
Effects of five single factors on DPPH radical scavenging activity. (**A**) Enzyme concentration (U/g); (**B**) solid:liquid ratio (*w*/*v*); (**C**) hydrolysis time (h); (**D**) pH; and (**E**) temperature (°C). The concentration of hydrolyzed product was 5 mg/mL and 10 mg/mL, respectively. All results were triplicates of the mean ± SD.

**Figure 3 marinedrugs-18-00430-f003:**
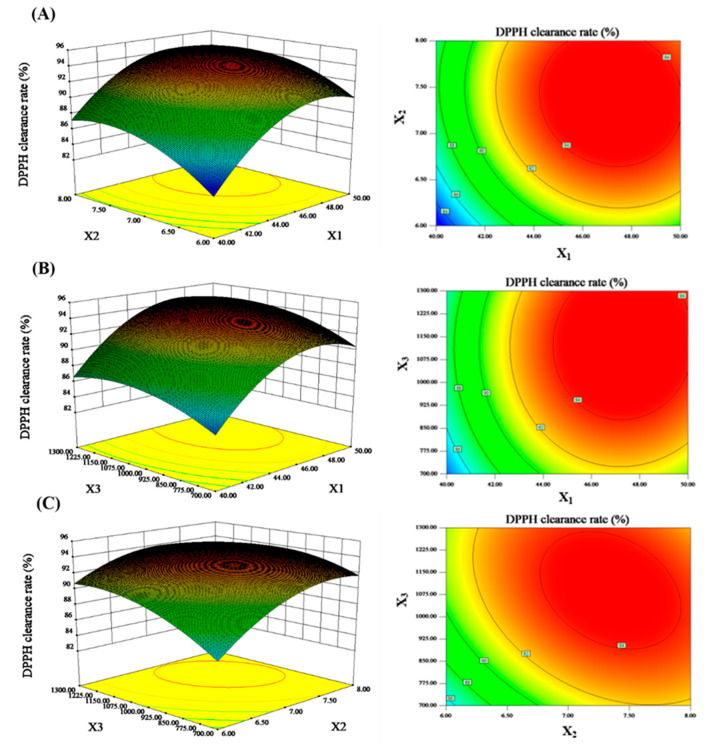
Response surface plots showing the effects of different variables. Images on the left represent three-dimensional response surface plots, whereas images on the right represent two-dimensional contour plots. Images represent the following: (**A**) Effects of temperature (X1) and pH (X2) on the DPPH radical scavenging activity; (**B**) Effects of temperature (X1) and enzyme concentration (X3) on the DPPH radical scavenging activity; and (**C**) Effects of pH (X2) and enzyme concentration (X3) on the DPPH clearance rate.

**Figure 4 marinedrugs-18-00430-f004:**
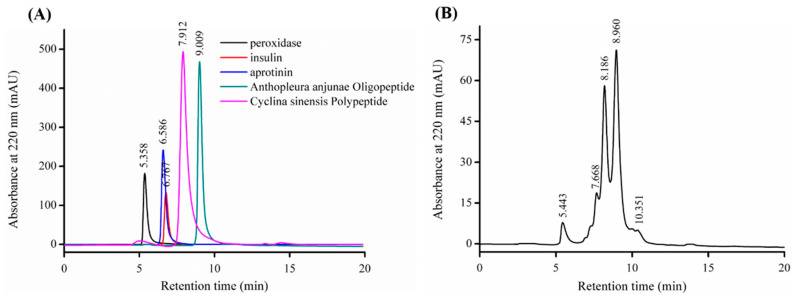
(**A**) The HPLC chromatograms of the standard molecular weight samples; (**B**) the molecular weight distribution of SNNHs from the swim bladders of *Nibea japonica*. (Mobile phase: water/acetonitrile/trifluoroacetic acid = 55:45:0.1, flow rate: 0.5 mL/min.).

**Figure 5 marinedrugs-18-00430-f005:**
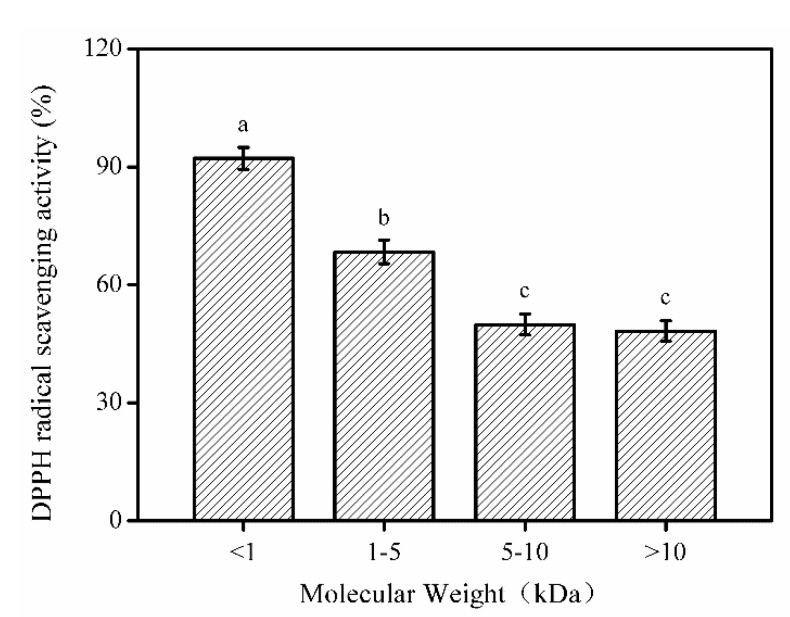
The effect of four fractions of SNNHs on DPPH radical scavenging activity. Results were triplicates of the mean ± SD. Different letters indicate significant differences between groups (*p* < 0.05).

**Figure 6 marinedrugs-18-00430-f006:**
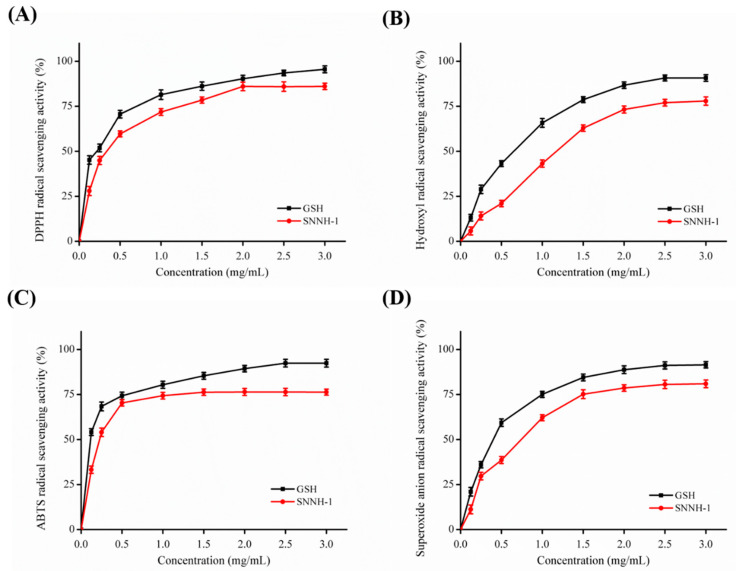
DPPH radical (**A**), hydroxyl radical (**B**), ABTS radical (**C**), and superoxide anion radical (**D**) scavenging activities of SNNH-1. All results were triplicates of the mean ± SD.

**Figure 7 marinedrugs-18-00430-f007:**
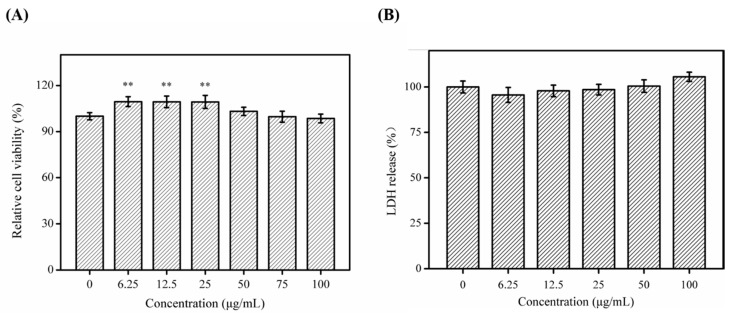
Effects of treatment with different concentrations of SNNH-1 for 24 h on relative cell viability (**A**) and LDH (**B**) in HUVECs. Data are presented as the mean ± SD (*n* = 6). ** *p* < 0.01 vs. the Control group.

**Figure 8 marinedrugs-18-00430-f008:**
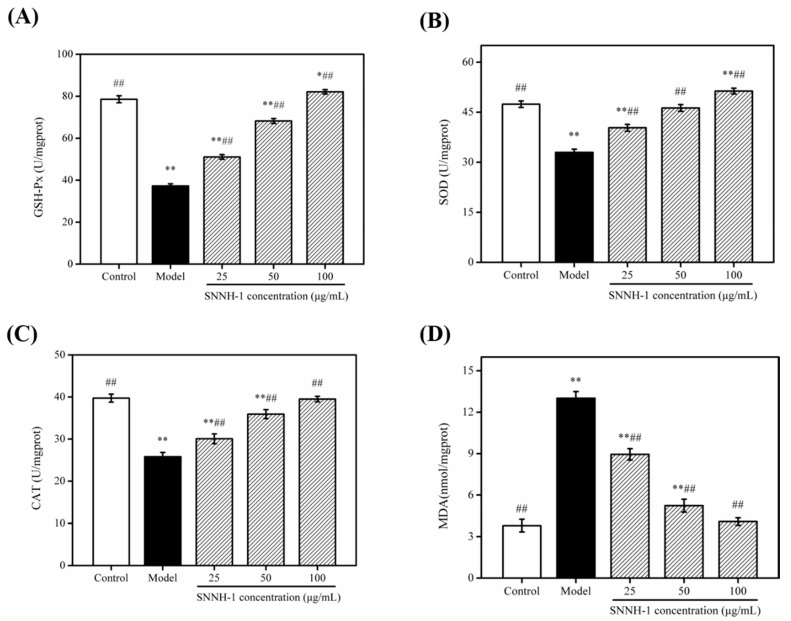
Effect of SNNH-1 at different concentrations (25, 50, and 100 µg/mL) on levels of GSH-Px (**A**), SOD (**B**), CAT (**C**), and MDA (**D**) in an H_2_O_2_-induced HUVECs injury model. Data are presented as the mean ± SD (*n* = 6). * *p* < 0.05, ** *p* < 0.01 vs. the Control group, ^##^
*p* < 0.01 vs. the Model group (treatment with 600 µM H_2_O_2_).

**Figure 9 marinedrugs-18-00430-f009:**
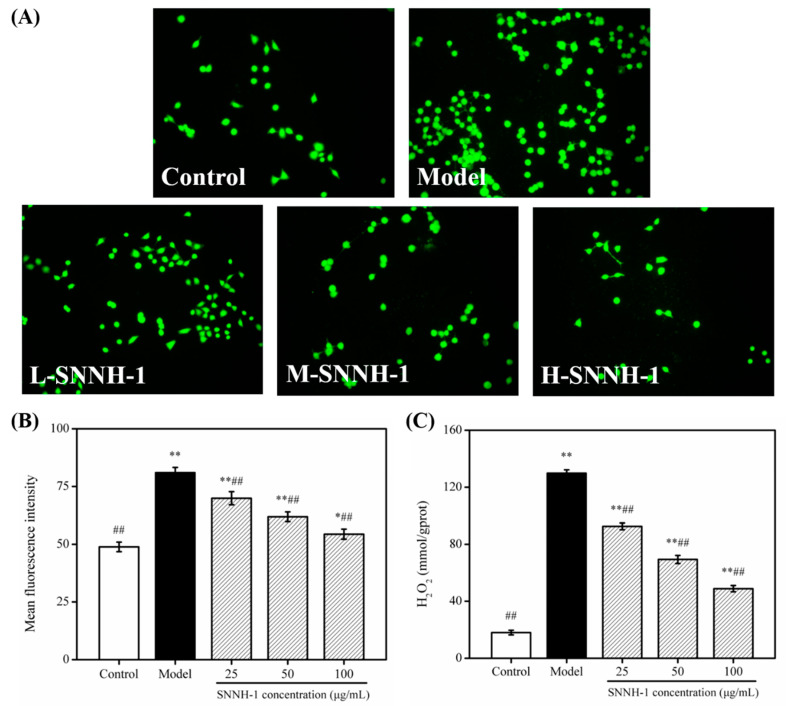
Effect of SNNH-1 on ROS levels in an H_2_O_2_-induced HUVECs injury model. (**A**) Fluorescence images of different groups observed by a fluorescence microscope; (**B**) mean fluorescence intensity of cells in different groups; (**C**) H_2_O_2_ levels in HUVECs of different groups. Data are presented as the mean ± SD (*n* = 3). * *p* < 0.05, ** *p* < 0.01 vs. the Control group, ^##^
*p* < 0.01 vs. the Model group (treatment with 600 µM H_2_O_2_).

**Table 1 marinedrugs-18-00430-t001:** The Box–Behnken design and the response for the DPPH clearance rate.

Runs	Temperature (*X*_1_)	pH (*X*_2_)	Enzyme Concentration (*X*_1_)	DPPH Clearance Rate (%) (*Y*)
1	40	7	700	84.43
2	50	7	1300	93.24
3	45	8	1300	93.09
4	45	7	1000	94.33
5	45	6	1300	90.92
6	50	7	700	91.03
7	45	7	1000	93.78
8	40	8	1000	86.98
9	45	6	700	84.61
10	50	8	1000	92.62
11	40	7	1300	86.06
12	50	6	1000	90.24
13	45	8	700	91.51
14	45	7	1000	94.36
15	40	6	1000	82.71

**Table 2 marinedrugs-18-00430-t002:** Analysis of variance of the regression model.

Source	Sum of Squares	*df*	Mean Square	*F Value*	*p Value*
Model	218.02	9	24.22	39.82	0.0004
*X* _1_	90.79	1	90.79	149.22	<0.0001
*X* _2_	30.89	1	30.89	50.77	0.0008
*X* _3_	17.20	1	17.20	28.27	0.0031
*X* _1_ *X* _2_	0.89	1	0.89	1.47	0.2798
*X* _1_ *X* _3_	0.084	1	0.084	0.14	0.7253
*X* _2_ *X* _3_	5.59	1	5.59	9.19	0.0290
*X* _1_ ^2^	50.03	1	50.03	82.22	0.0003
*X* _2_ ^2^	20.19	1	20.19	33.18	0.0022
*X* _3_ ^2^	11.78	1	11.78	19.35	0.0070
Residual	3.04	5	0.61		
Lack of fit	2.83	3	0.94	8.84	0.1033
Pure Error	0.21	2	0.11		
Cor Total	221.07	14			
*R* ^2^					0.9862
*R_adj_* ^2^					0.9615

**Table 3 marinedrugs-18-00430-t003:** Composition and contents of amino acids of SNNH-1.

Amino Acid	SNNH-1
Aspartic acid	38
Threonine *	14
Serine	24
Glutamic acid	76
Glycine	340
Alanine	134
Cysteine	0
Valine *	18
Methionine *	14
Isoleucine *	8
Leucine *	23
Tyrosine	7
Phenylalanine *	16
Histidine	6
Lysine *	28
Arginine	59
Proline	105
Hydroxyproline	86
Imino acid	191

* Human-essential amino acids.

**Table 4 marinedrugs-18-00430-t004:** Factors and levels in the response surface design.

Independent Factors	Symbol	Level of Factor		
		−1	0	1
Temperature (°C)	X_1_	40	45	50
pH	X_2_	6	7	8
Enzyme concentration (U/g)	X_3_	700	1000	1300

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
