# Peer review of "Collagen Peptides from Swim Bladders of Giant Croaker (Nibea japonica) and Their Protective Effects against H2O2-Induced Oxidative Damage toward Human Umbilical Vein Endothelial Cells"

_marinedrugs, 2020, doi:10.3390/md18080430_

Round 1

Reviewer 1 Report

The description of the work, as well as, introduction and results/discussion should be more clear and concise. The sentences are confusing and sometimes it is difficult to understand them. Please, read again some suggestions and try to be more scientific. Words such as "in addition". "example" are often written in the manuscript

Author Response

Reviewer #1:

We thank the reviewer for valuable comments on our manuscript. As the reviewer pointed out, we made our effort to have this manuscript much better. We have revised the manuscript accordingly.

Specifically:

  1. The description of the work, as well as, introduction and results/discussion should be more clear and concise. The sentences are confusing and sometimes it is difficult to understand them. Please, read again some suggestions and try to be more scientific. Words such as "in addition". "example" are often written in the manuscript.

Re: We read the manuscript carefully and revised it based on the reviewers’ suggestions. We also asked professional English editing services to modify some typing and grammatical mistakes to improve our manuscript.

Reviewer 2 Report

The manuscript describes that the some functions of collagen peptides, produced by several methods, from swim bladders of giant croaker.

It might be valuable for engineering fields, but not be interested in a marine biosciences.

The results looks similar to their previous reports.

It should be published on other journal of bio-engineering.

Author Response

  1. The manuscript describes that the some functions of collagen peptides, produced by several methods, from swim bladders of giant croaker.It might be valuable for engineering fields, but not be interested in a marine biosciences. The results looks similar to their previous reports. It should be published on other journal of bio-engineering.

Re: In the previous study, we extracted the collagen from swim bladders of giant croaker (Nibea japonica). Some properties of this collagen were studied, and we also evaluated the wound healing ability of collagen sponge. The purpose of the previous study was aimed to use the collagen sponge to promote the wound healing process. (Physicochemical, antioxidant properties of giant croaker (Nibea japonica) swim bladders collagen and wound healing evaluation. International Journal of Biological Macromolecules 138 (2019) 483–491).

However, in the present study, we aimed to prepare the antioxidant collagen peptide from the swim bladders of Nibea japonica, and wanted to use these collagen peptides in the food industry, pharmaceuticals, and cosmetics. In this study, the extraction conditions were optimized and collagen peptides with a Mw of less than 1 kDa (SNNH-1) was were obtained by ultrafiltration, and exhibited good scavenging activity for hydroxyl radicals, ABTS radicals and superoxide anion radicals. We also found that SNNH-1 has the protective effect against oxidative damage of H2O2-induced HUVECs.

Reviewer 3 Report

Work by Zheng and co-authors “Collagen Peptides from Swim Bladders of Giant Croaker (Nibea japonica) and Their Protective Effects against H2O2-induced Oxidative Damage toward Human Umbilical Vein Endothelial Cells” describes anti-oxidant properties of the newly identified peptide purified from Nibea japonica. It is an interesting study that addresses the important question in the field of redox biology on novel ROS scavengers and their potential practical application. However, this manuscript requires significant textual changes and at least one experimental evidence prior publication in the Journal of Marine Drugs.

  1. All figure legends must be re-written and provide detailed explanation on how experiments were done, indicate what each image represents and describe statistical analysis details where applicable.
  2. The first experimental section 2.1.1.”Selection of the Optimal enzyme” is confusing. I still do not understand what the authors were trying to achieve and a rational behind the experiment described. Same critique is for the following section 2.1.2
  3. Results presented on Figure 6 compare the antioxidant properties of SNNH-1 peptide and ascorbic acid that the author used as a known ROS scavenger. However, under certain conditions ascorbic acid might play a role of the pro-oxidant. Therefore, the choice of positive control is unfortunate. This experiment should be repeated with different control, such as reduced glutathione GSH.

4.Figure 8: how activity of CAT, SOD and GSH-Px were determined? There is no information about assays used neither in Figure legend, nor in Materials and methods or Results. Scientific description of the performed experiments must be provided;

  1. Figure 9: Authors present images of fluorescent cells after different treatment. Where the fluorescent signal is coming from? How many cells were analyzed? Where is the statistics? All these questions must be addressed. I also recommend to use Amplex red assay to estimate the concentration of H2O2.

Author Response

Reviewer #3:

We thank the reviewer for valuable comments on our manuscript, which have made our manuscript much better. We have revised the manuscript according these comments.

Specifically:

  1. All figure legends must be re-written and provide detailed explanation on how experiments were done, indicate what each image represents and describe statistical analysis details where applicable.

Re: We have re-written all the figure legends and added relevant experimental methods and details.

  1. The first experimental section 2.1.1. “Selection of the Optimal enzyme” is confusing. I still do not understand what the authors were trying to achieve and a rational behind the experiment described. Same critique is for the following section 2.1.2.

Re: The five proteases used in this study differed in the selectivity of restriction sites. The molecular weight and amino acid residues of the peptides obtained by the enzyme digestion are different depending on the site of the protease digestion. Correspondingly, the antioxidant activity in vitro is also different. In order to obtain peptides with the best antioxidant activity, the enzymes were screened in this study. Similarly, the determination of the single-factor optimal conditions is also to obtain the peptide with the best antioxidant activity.

  1. Results presented on Figure 6 compare the antioxidant properties of SNNH-1 peptide and ascorbic acid that the author used as a known ROS scavenger. However, under certain conditions ascorbic acid might play a role of the pro-oxidant. Therefore, the choice of positive control is unfortunate. This experiment should be repeated with different control, such as reduced glutathione GSH.

Re: We have selected GSH as a positive control and retested the antioxidant properties of SNNH-1 in vitro.

  1. Figure 8: how activity of CAT, SOD and GSH-Px were determined? There is no information about assays used neither in Figure legend, nor in Materials and methods or Results. Scientific description of the performed experiments must be provided.

Re: We added information about experimental method in section 3.8 of the revised manuscript: HUVECs were inoculated into 6-well plates (1×105 cells/well) and incubated in a 5% CO2 incubator at 37 °C for 24 h. SNNH-1 at final concentrations of 0, 25, 50 and 100 µg/mL were added into the protection groups and cultured for another 24 hours. Each group was treated with 600 µmol/L H2O2 for 4 h. The group treated without SNNH-1 and H2O2 was used as the control group. Subsequently, 500 μL of cell lysis buffer was added to each well on ice. Cells were lysed for 30 min, and centrifuged at 12,000 rpm for 10 min at 4 °C. The resulting supernatant was stored at 4 °C. Levels of GSH-Px, SOD, CAT, MDA, and H2O2 were determined by using assay kits according to the manufacturers’ instructions (Nanjing Jiancheng Bioengineering Institute, Nanjing, China), and protein concentrations were determined using the bicinchoninic acid (BCA) method.

  1. Figure 9: Authors present images of fluorescent cells after different treatment. Where the fluorescent signal is coming from? How many cells were analyzed? Where is the statistics? All these questions must be addressed. I also recommend to use Amplex red assay to estimate the concentration of H2O2.

Re: We added information about experimental method in section 3.9 of the revised manuscript: HUVECs were seeded in 6-well plates with a density of 1×104 cells/mL for 24 h. SNNH-1 at final concentrations of 25, 50 and 100 µg/mL was added to the protection groups 24 h and HUVECs were treated with 600 µmol/L H2O2 for 4 h in a 5% CO2 incubator at 37 °C. Production of ROS was determined using a ROS Assay Kit (Nanjing Jiancheng Bioengineering Institute, Nanjing, China). Next, a total of 2 mL of 2’,7’-dichlorodihydrofluorescein diacetate (DCFH-DA) fluorescent probe solution (10 µM) was added to the cells, and incubated for 30 min. Then cells were washed three times with serum-free Dulbecco’s modified Eagle medium (DMEM) medium and observed under a fluorescence microscope (Axio Imager A2, Carl Zeiss, Germany). The fluorescence intensity was analyzed using Image J software.

In addition, we detected the level of H2O2 in the HUVECs, the results were shown in Figure 9C.

Round 2

Reviewer 2 Report

Many collagen related studies have already reported. If authors would like to add new scientific insights in this field, they have to add data of some control collagen. Usually, pig derived collagen was used as a standard. On the other hands, a collagen tripeptide, GHK, was used for a minimal unit of collagen.

If those standards were used in the study, readers could recognise the effects of the low molecular weight fish collagen.

I could not evaluate the effects of the low molecular weight fish collagen from the current data.

Author Response

We thank the reviewer for valuable comments on our manuscript. As the reviewer pointed out, we made our effort to have this manuscript much better. We have revised the manuscript accordingly.

Specifically:

  1. Many collagen related studies have already reported. If authors would like to add new scientific insights in this field, they have to add data of some control collagen. Usually, pig derived collagen was used as a standard. On the other hands, a collagen tripeptide, GHK, was used for a minimal unit of collagen.

If those standards were used in the study, readers could recognize the effects of the low molecular weight fish collagen.

I could not evaluate the effects of the low molecular weight fish collagen from the current data.

Re: Thanks for your comment. Your comment is very professional and correct. We regret that we did not take this into consideration when designing the experiment. However, as the other reviewers’ suggestion, we assessed the antioxidant activity of SNNH-1 and used the glutathione (GSH) as an activity control. The antioxidant activity of SNNH-1 is close to that of GSH. Therefore, the high free radical scavenging activity of SNNH-1 indicates that it is a potential candidate to be developed as an antioxidant and can be used in anti-aging health products and cosmetics. Furthermore, in our manuscript, we also compared the cytoprotective function of SNNH-1 from the swim bladders of Nibea japonica with the low molecular weight collagen peptide from the Nibea japonica skin and sea cucumbers (Acaudina molpadioides), our results were similar with these previous studies.

Reviewer 3 Report

The authors have improved their manuscript. I have no hesitation to recommend acceptance. 

Author Response

Thank you for your suggestion.